# Boundary-Attentive 3D-UNet for Auto-Segmentation of Tumor-Prone Organs in Medical CT Volumes

**Obinna Agbodike**[*1] iD                                    OBINNADYKE@GMAIL.COM
**Chang-Fu Kuo**[1,2,3] iD                                    ZANDIS@GMAIL.COM

[1] *Center for Artificial Intelligence in Medicine, Chang Gung Memorial Hospital, Taoyuan, Taiwan*
[2] *Department of Internal Medicine, College of Medicine, Chang Gung University, Taoyuan, Taiwan*
[3] *Division of Rheumatology, Allergy and Immunology, Linkou Chang Gung Memorial Hospital, Taoyuan, Taiwan*

## Abstract

Advances in deep learning and the evolution of neural architectures continue to push the limits of organ and tumor-level segmentation fidelity in complex CT environments. Building on this momentum, we introduce a Boundary-Attentive 3D-UNet (BAUN3D), a unique tumor-centric and structure-learning framework designed for the supervised segmentation of organs prone to tumor occurrence in volumetric CT images. BAUN3D incorporates a gated boundary-refinement module that enhances local organ-tumor interfaces while preserving global organ geometry. Trained from scratch with a composite Dice, Focal-Tversky, Tumor, and Boundary loss formulated to support contour structural continuity, the model achieves competitive performance on two major Medical Segmentation Decathlon (MSD) benchmarks, attaining organ/tumor Dice scores of $\approx 0.95/0.71$ on LiTS and $\approx 0.91/0.78$ on the more challenging Pancreas dataset, respectively. Overall, the results highlight the validity of boundary-context modeling for anatomical and anomaly segmentation in volumetric CT. Code is available at: https://github.com/obinnadyke/baun3d

**Keywords:** Deformable Attention, CT, Liver, Pancreas, Segmentation, Tumor, UNet

## 1. Introduction

Accurate and reliable segmentation of human parenchymal organs, particularly those susceptible to tumor incidence such as the liver (hepatocellular carcinoma) and pancreas (adenocarcinoma), is paramount in oncologic imaging, disease assessment, and radiotherapy planning (Azad et al., 2024; Antonelli et al., 2022). However, achieving consistent volumetric organ–tumor delineation, especially in abdominal computed tomography (CT) remains challenging (Hu et al., 2025). Scanner-dependent variations, heterogeneous tumor appearance, partial-volume effects, and inherent noisy or low inter-tissue contrast often obfuscate subtle anatomical boundaries, making accurate delineation of overlapping structures difficult for both expert annotators and automated radiomics systems.

Although many recent UNet and vision transformer derivatives (Isensee et al., 2018; Hatamizadeh et al., 2022; Azad et al., 2024; Kareem et al., 2024; Agbodike et al., 2025) report strong Dice coefficients across public benchmark datasets, qualitative deviations

---

[*] Corresponding author

occurring around organ–tumor interfaces remain common. Such errors, whether arising from annotation inconsistencies in non-expert labeling workflows (Karimi et al., 2020; Gurari et al., 2015; Zhou et al., 2025) or from limitations of the underlying deep neural network (DNN) architectures (Ram et al., 2025), can degrade segmentation fidelity especially in the anatomical regions where clinical decisions depend most on accurate boundary definition.

Motivated by these concerns, this work presents a boundary-attentive DNN (dubbed BAUN3D), designed to stabilize the geometric localization and automatic contour delineation of organ–tumor texture representations in medical CT volumes.

## 2. Methodology

The BAUN3D architecture follows the standard 3D-UNet encoder-decoder topology (Çiçek Özgün et al., 2016), whereby an input CT volume $x \in \mathbb{R}^{1 \times D \times H \times W}$ is processed through (four) residual encoding stages $s$ to extract hierarchical features $\{x_1, x_2, x_3, x_4\}$. The key innovation, however, lies in the symmetric decoder pathway, which integrates novel boundary-attentive mechanisms that refines organ–tumor interfaces during the reconstruction of the segmentation probability map $\hat{y} \in [0, 1]^{C \times D \times H \times W}$.

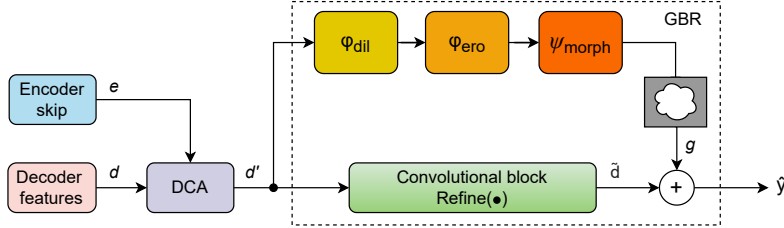

Figure 1: Overview of the boundary refinement pipeline

### 2.1. The Boundary-Refinement Strategy

At each decoder stage $s \in \{1, \ldots, 3\}$, we first apply deformable cross-attention (DCA) to adaptively fuse encoder skip features $e$ with decoder features $d$. Using linear projection weights $W_a$ (attention), $W_v$ (value), and $W_p$ (post-attention mixing), the DCA computes feature fusion $\alpha = \mathrm{softmax}(W_a[e|d]/\tau)$ to produce enhanced representations $d' = d + W_p(\alpha \odot W_v e)$, where $\tau$ is a temperature parameter that controls the selectivity of foreground high-frequency responses. To restore the structural boundary detail, $d'$ is passed through a gated boundary refinement (GBR) module that aggregates multi-scale morphological cues:

$$f = [\, d', \, \phi_{\mathrm{dil}}(d'), \, \phi_{\mathrm{ero}}(d'), \, \psi_{\mathrm{morph}}(d') \,], \qquad g = \sigma(W_f f), \qquad r = \mathrm{Refine}(d' \odot g),$$

where $\phi_{\mathrm{dil}}$, $\phi_{\mathrm{ero}}$, and $\psi_{\mathrm{morph}}$ denote dilation, erosion, and morphology-based gradient operators; $\sigma$ is the sigmoid activation of attention mask $g$, and $\mathrm{Refine}(\cdot)$ is a convolutional refinement block. The final boundary-enhanced feature is given by $\tilde{d} = d' + \gamma r$, where $\gamma = \sigma(\mathrm{gate})$ is a learnable scalar that regulates contribution of the boundary corrections. A flowchart of these processes is illustrated in Fig. 1.

## 3. Implementations and Results

### 3.1. Loss Function

The model training was based on a supervised learning approach optimized with an objective composite loss $\mathcal{L} = \lambda\mathcal{L}_{\mathrm{D}} + \lambda\mathcal{L}_{\mathrm{F}} + \lambda\mathcal{L}_{\mathrm{T}} + \lambda\mathcal{L}_{\mathrm{B}}$, where Dice loss $\mathcal{L}_{\mathrm{D}}$ stabilizes volumetric overlap, the Focal–Tversky term $\mathcal{L}_{\mathrm{F}}$ manages data class imbalance, the tumor-focused component $\mathcal{L}_{\mathrm{T}}$ amplifies gradients for small lesions, and the boundary loss $\mathcal{L}_{\mathrm{B}}$ encourages alignment with signed-distance maps. Each term is weighted by a class-aware coefficient $\lambda$ and implemented with numerically stable empty-mask handling.

### 3.2. Experimental Settings

Experiments were performed on the LiTS (Task 03) and Pancreas (Task 07) datasets following the official Medical Segmentation Decathlon protocol (Antonelli et al., 2022). Input intensities were clipped to $[-100, 400]$ HU for LiTS and $[-100, 350]$ HU for Pancreas. Learning rates were set to $2 \times 10^{-4}$ for LiTS and $1 \times 10^{-4}$ for Pancreas. Both tasks employed the AdamW optimizer, a cosine learning-rate scheduler with warm-up, a 70/30 train–validation split, and 450 training epochs. All experiments were implemented in PyTorch and executed with two NVIDIA RTX 3080 GPUs using data-parallel for batch sizes $\geq 2$.

### 3.3. Summary of Outcomes

Table 1 and Table 2 report quantitative results comparing the BAUN3D with selected state-of-the-art DNN baselines on the LiTS (liver and tumor) and Pancreas (organ and tumor) datasets, respectively. Evaluation metrics include per-class Dice coefficients, and average 95% Hausdorff Distance (HD95). Overall, the BAUN3D achieves competitive performance, particularly with strong tumor Dice improvement on the more challenging Pancreas task.

| Method | Year | Liver Dice↑ | Tumor Dice↑ | Avg. Dice↑ | Avg. HD95%↓ |
| --- | --- | --- | --- | --- | --- |
| nn-UNet 3D Cascade | (2018) | 0.95 | 0.62 | 0.79 | − |
| Swin UNETR | (2022) | 0.87 | 0.52 | 0.70 | 31.19 |
| vMixer | (2024) | 0.95 | 0.58 | 0.78 | 19.48 |
| BAUN3D | (Ours) | 0.95 | **0.71** | **0.83** | **10.78** |

Table 1: Quantitative results on the MSD LiTS (Liver-Tumor) dataset $\{n = 131\}$.

| Method | Year | Panc. Dice↑ | Tumor Dice↑ | Avg. Dice↑ | Avg. HD95%↓ |
| --- | --- | --- | --- | --- | --- |
| nn-UNet 3D Cascade | (2018) | 0.79 | 0.49 | 0.64 | − |
| Swin UNETR | (2022) | 0.72 | 0.36 | 0.55 | 18.24 |
| vMixer | (2024) | 0.80 | 0.49 | 0.65 | **5.85** |
| BAUN3D | (Ours) | **0.91** | **0.78** | **0.84** | 7.55 |

Table 2: Quantitative results on the MSD Pancreas (Organ-Tumor) dataset $\{n = 281\}$.

In Appendix A, the qualitative results for representative liver–tumor and pancreas–tumor multi-view cases are provided in Fig. 2 and Fig. 3, respectively, showing the correspondence between ground-truth annotations and BAUN3D predictions, alongside attention heatmaps demonstrating the geometric boundary-refinement behavior of the model.

## Acknowledgments

This work was supported by funding from the Chang Gung Memorial Hospital Research Project, under grant no. CLRPG3H0017.

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

## Appendix A. Qualitative Results

### A.1. Liver and Tumor Learning Outcome

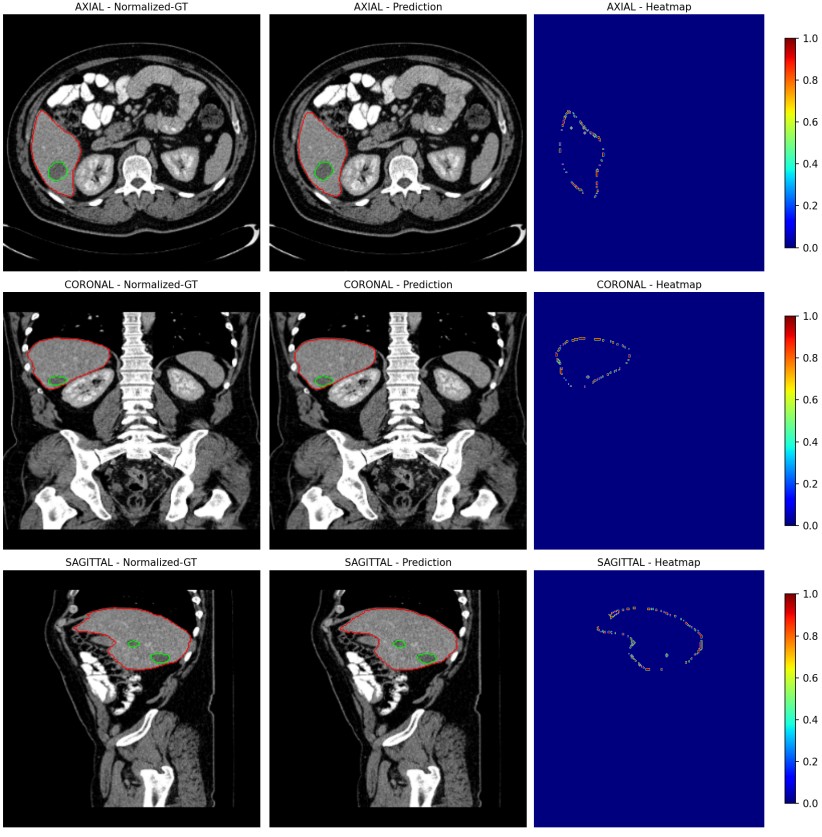

Figure 2: Multi-view liver–tumor segmentation (Organ:Red, Tumor:Green).

## A.2. Pancreas and Tumor Learning Outcome

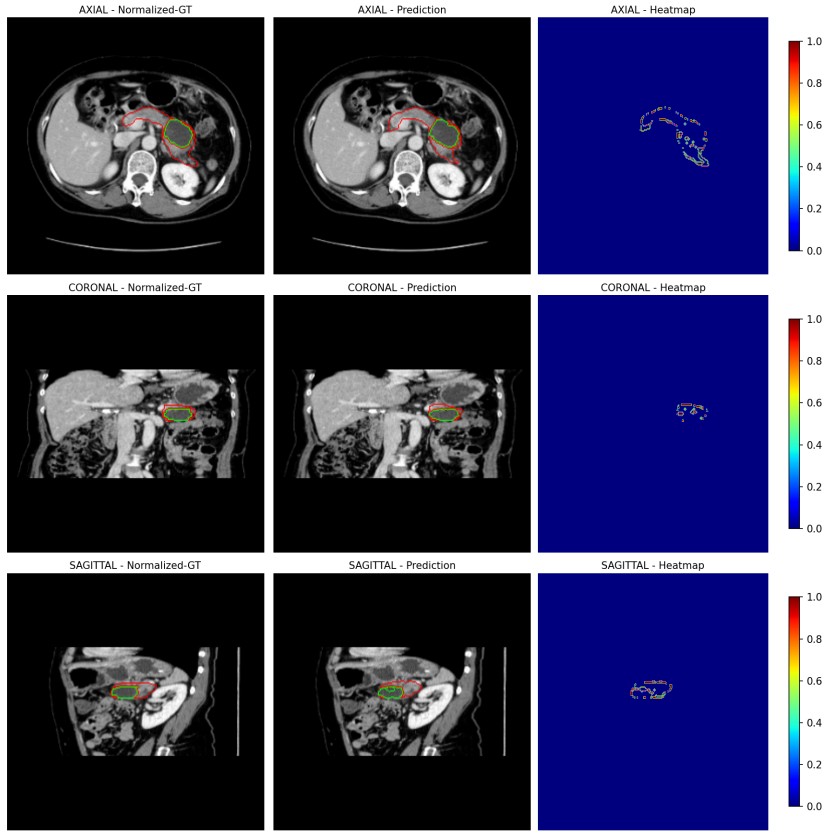

Figure 3: Multi-view pancreas–tumor segmentation (Organ:Red, Tumor:Green).

