# OpenReview forum: "Boundary-Attentive 3D-UNet for Auto-Segmentation of Tumor-Prone Organs in Medical CT Volumes"
_MIDL.io/2026/Short_Papers — MIDL 2026 - Short Papers Poster_

### Official Review · Reviewer_HkhW · 2026-05-06
**Solid idea on boundary-aware segmentation, but needs clearer validation and stronger novelty positioning**

**Rating:** 3
**Confidence:** 4

**Review:**

- The paper addresses an important and well-known challenge in medical image segmentation: accurately capturing organ–tumor boundaries, which are often the most clinically relevant yet hardest regions to model. The idea of explicitly modeling boundary information within a 3D UNet framework is reasonable and aligns with trends in recent literature.
- From a technical standpoint, the combination of deformable cross-attention and the gated boundary refinement (GBR) module is interesting, particularly the use of morphological operators (dilation/erosion) within the feature space. This adds an interpretable component to the architecture and is a nice design choice. The composite loss is also well-motivated and aligns with the goal of improving small tumor segmentation and boundary precision.
- However, the novelty feels somewhat incremental. There is a large body of work on boundary-aware losses, attention mechanisms, and UNet variants, and the paper does not clearly position how this approach is meaningfully different or better beyond combining known components. The contribution would benefit from a sharper ablation study isolating the effect of each component (DCA, GBR, boundary loss). Right now, it is hard to attribute the gains to specific design choices.
- On the experimental side, results are promising, especially the tumor Dice improvement on the pancreas dataset. That said, the evaluation is somewhat limited. Comparisons are made against a small set of baselines, and it is unclear whether these are reproduced under the same conditions or taken from prior work. Also, nnU-Net is shown with relatively low tumor Dice compared to typical reported numbers, which raises questions about fairness of comparison.

**Summary:**

This paper proposes BAUN3D, a boundary-attentive extension of 3D-UNet designed to improve segmentation of tumor-prone organs in CT volumes, specifically liver and pancreas. The method integrates deformable cross-attention with a gated boundary refinement module that leverages morphological operations (dilation, erosion, gradients) to enhance organ–tumor interfaces. The model is trained with a composite loss combining Dice, Focal-Tversky, tumor-specific, and boundary-aware terms. Experiments on MSD LiTS and Pancreas datasets show competitive performance, with noticeable gains in tumor Dice, especially on pancreas. Overall, the work emphasizes the importance of boundary modeling for improving segmentation fidelity in challenging regions.

**Strengths:**

- Targets a highly relevant problem: improving organ–tumor boundary delineation, which is critical in clinical workflows.
- The boundary refinement module is intuitive and incorporates morphological priors in a meaningful way.
- Combination of attention and boundary modeling is well-motivated.
- Shows strong empirical performance, especially for tumor segmentation on pancreas (a challenging task).
- Uses multiple loss components that align well with the problem (class imbalance + boundary sensitivity).
- Code availability is a strong plus for reproducibility.

**Weaknesses:**

- Novelty is somewhat limited; the method mainly combines existing ideas (UNet + attention + boundary loss + morphology).
- Lack of ablation studies makes it hard to understand which components actually drive performance gains.
- Baseline comparisons are limited and potentially not fully fair or consistent.
- No dedicated boundary-specific evaluation metrics, despite boundary being the core claim, also I found limited discussion of generalization (e.g., cross-dataset performance, robustness).
- Some clarity issues in method description and notation; could be simplified, also the results seems strong but are not clearly beyond  SOTA across all settings.

**Justification Of Rating:**

The paper presents a reasonable and well-motivated approach with solid empirical results, particularly on challenging tumor segmentation tasks. However, the contribution feels incremental, and the evaluation does not fully support the central claims around boundary improvement. With stronger ablations, clearer positioning against prior work, and more targeted evaluation metrics, this could move toward accept. In its current form, it sits around the borderline.

---

### Decision · Program_Chairs · 2026-05-08

Accept (Poster)